# Mechanical Properties of Polypropylene: Additive Manufacturing by Multi Jet Fusion Technology

**DOI:** 10.3390/ma14092165

**Published:** 2021-04-23

**Authors:** Jiří Šafka, Michal Ackermann, Filip Véle, Jakub Macháček, Petr Henyš

**Affiliations:** 1The Institute for Nanomaterials, Advanced Technologies and Innovation, Technical University of Liberec, Studentská 1402/2, 461 17 Liberec, Czech Republic; michal.ackermann@tul.cz; 2Faculty of Mechanical Engineering, Technical University of Liberec, Studentská 1402/2, 461 17 Liberec, Czech Republic; filip.vele@tul.cz; 3Faculty of Mechatronics, Informatics and Interdisciplinary Studies, Technical University of Liberec, Studentská 1402/2, 461 17 Liberec, Czech Republic; jakub.machacek@tul.cz (J.M.); petr.henys@tul.cz (P.H.)

**Keywords:** additive manufacturing, polypropylene, powder recycling, mechanical properties, SEM

## Abstract

Multi jet fusion (MJF) technology has proven its significance in recent years as this technology has continually increased its market share. Recently, polypropylene (PP) was introduced by Hewlett-Packard for the given technology. To our knowledge, little is known about the mechanical properties of polypropylene processed by MJF technology. During this study, standardised specimens were printed under all of the major orientations of the machine’s build space. Each of these orientations were represented by five samples. The specimens then underwent tensile, bending and Charpy impact tests to analyse their mechanical properties. The structural analysis was conducted to determine whether PP powder may be reused within the MJF process. The mechanical tests showed that the orientation of the samples significantly influences their mechanical response and must be carefully chosen to obtain the optimal mechanical properties of PP samples. We further showed that PP powder may be reused as the MJF process does not significantly alter its thermal and structural properties.

## 1. Introduction

The multi jet fusion (MJF) method was first introduced by Hewlett-Packard in 2016. Thanks to its speed, overall printing quality and the affordable price of its products, MJF technology is often used in the segment of small series production of plastic parts [1]. In comparison with selective laser sintering (SLS), MJF produces more ductile samples with smoother surfaces [2,3]. The work of Tasch et al. [4] reports significant material inhomogeneities in PA12, possibly influencing crack initiation. Compared to other additive manufacturing (AM) technologies, MJF is more efficient in production and more ecological [5]. The first material was introduced for this technology was Polyamide 12 (PA12). Nowadays, this technology is able to produce models from Polyamide 12 with glass beads (PA12GB), thermoplastic polyurethane (TPU) and since 2020, polypropylene (PP).

Polypropylene (PP) is a semi-crystalline thermoplastic material used as a (nano)composite [6,7,8] and biomimetic material [9,10,11]. It is a material suitable for corrosion-resistant applications [12,13,14] and it is biocompatible [15]. Regarding currently available AM techniques, PP is predominately produced by fused deposition modelling (FDM) technology and it is frequently used as a phase in composites. [16,17,18,19]. The main drawbacks of PP include its tendency to warp during the FDM process [20] and its low adhesion to a substrate plate [18,21,22]. SLS is also used for manufacturing PP materials [13,23,24,25]. Compared to the injection moulding (IM), SLS manufactured PP parts are less ductile [13].

The alternation of mechanical properties in relation to the orientation of the part is a well-known phenomenon in the field of additive manufacturing. For some technologies, such as the FDM, a considerable decrease in tensile strength may be observed if the loads act in a direction normal to the layering surface of the product [26]. MJF technology, on the other hand, offers a product with a more compact structure due to its unique layer processing approach. As a result, the anisotropy of the material is less obvious than for other AM technologies. Lee et al. [27] studied the influence of build orientation for PA11 material processed using MJF technology. Their results showed that the main differences are in the strain at break and the modulus of elasticity rather than tensile strength. Similar findings may be found in the work of O’Connor et al. [28] which deals with an evaluation of the mechanical response for PA12 material.

Knowledge of the mechanical properties of PP produced by AM is rather limited, and to the authors’ knowledge, a combination of the PP and MJF methods has never been considered even though it may provide a powerful combination for material and engineering design applications. In this study, the authors attempted to describe the mechanical response of PP produced by MJF under various different sample orientations in the MJF process. Suitable printing orientation combinations were selected as the authors believed that the orientation significantly influences the resultant mechanical properties of the PP samples.

## 2. Materials and Methods

To fulfil the scope of this study, three types of mechanical tests were selected. Specifically, the tensile test, the three-point bending test and the notch impact test were performed to obtain detailed information about the mechanical response of PP processed by MJF. The input material was initially examined to evaluate its qualitative measures in all stages of its lifespan. For this reason, a powder size analysis followed by TGA and DSC measurements were performed for new, used and mixed PP powder.

### 2.1. Powder Analysis

The PP powder from BASF 3D Printing Solutions GmbH (Heidelberg, Germany) has a bulk density of 0.34 g/cm^3^ (according to ASTM D1895). The characteristic size distribution for the used powder is shown in Figure 1 and in Table 1. The powder size was measured by the laser diffraction method and evaluated according to Mieho’s interpretation. Five samples were used to estimate the size distribution.

A scanning electron microscope (SEM) was used to show the morphology of the PP powders. The new, used and mixed powders were examined under a Carl ZEISS Ultra Plus microscope (Carl Zeiss AG, Oberkochen, Germany) equipped with a FEG cathode. This instrument is able to operate in a low vacuum with accelerating voltage in the range of 0.5–30 kV. Moreover, a charge compensator (controlled Nitrogen blowing onto the surface of the sample) was used. Thanks to this feature, it was possible to capture even the non–conductive PP particles without the need of metal plating. The microscope is equipped with detectors for recording secondary electrons (SE2 and InLens) for the mapping surface morphology and electrons that were reflected (AsB, EsB) to display the chemical contrast. The parameters which were used for showing the morphology of the particles are summarized in Table 2. The result of the SEM analysis is shown in Figure 2. From the qualitative point of view, no major changes were found in the morphology and particle size distribution throughout all the tested samples.

### 2.2. Thermal Properties of the PP Powder

The thermal stability of the different PP powders (new, used and mixed) was analysed according to ISO 11358 on a TGA 2 device (Mettler Toledo, Greifensee, Switzerland). The weight of the samples was 6–9 mg. The samples first underwent a linear thermal heating from 50 °C to 600 °C at a speed of 10 °C/min in a nitrogen inert atmosphere and consequently up to 800 °C in an oxidation atmosphere. With three replications of the test, the degradation temperature was estimated from the degradation curves. For all the three measured powders, the degradation temperature was in the range of 462–465 °C.

The crystallisation ability and overall level of crystallinity of the PP powder were evaluated by differential scanning calorimetry (DSC) with DSC1/700 device (Mettler Toledo, Greifensee, Switzerland). The samples had a weight of 4.5 mg and they first underwent linear thermal heating from 0 °C to 200 °C in an inert atmosphere and consequently isothermic heating for 3 min. The samples were subsequently cooled to 0° at a speed of 10 °C/min. This step was required to remove the thermal history of the loading. In the next step, the samples underwent heating from 0 °C to 200 °C according to ISO 11357. The crystallinity phase starts at 110 °C according to the DSC diagram (Figure 3). The melting point of the PP powder is 138 °C, see Figure 4. Similar to the TGA analysis, three replications of the DSC test were done.

### 2.3. Specimen Preparation

For the tensile test, an ASTM D638-14 (Standard Test Method for Tensile Properties of Plastics) specimen with a thickness of 4 mm was chosen. The two remaining tests were performed using the same shape of the specimen pursuant to the ISO 179-1:2010 (Plastics—Determination of Charpy impact properties—Part 1: Non-instrumented impact test) standard. The geometry of the tensile test specimen and basic specimen shape for the bending and the impact tests are shown in Figure 5.

As soon as the shape of the specimens was defined, it was possible to prepare the build job. As noted in Section 1, a different orientation of the specimen in the build volume of the machine frequently leads to a shift in the mechanical response of the material. In order to test this phenomena for a combination of polypropylene material and MJF technology, multiple specimen positions were defined for the fabrication and further testing. The chosen orientations reflect the axes of the machine defined by the direction of the print head movement (X axis), the direction of the recoater movement (Y axis) and finally the travel of the substrate plate (Z axis), see Figure 6.

To distinguish all of the defined specimen orientations, a special marking system was used. The first part of the marking consisted of three digits, each of which may acquire a value of zero or one. These three digits mark the alignment of the specimen’s longitudinal axis with the X, Y and Z axes of machine’s build volume, respectively. The second part of the marking, separated from the first one by a hyphen, gives the angle value that defines the rotation of the specimen around its longitudinal axis. Within this part, 0°, 45° and 90° rotations were considered. This marking system is graphically expressed in Figure 7.

The figure shows that seven major orientations of the specimens were defined. Together with three different rotations within each of these directions, we obtained 21 unique specimens. Considering the planned statistical evaluation of the data, each combination was printed seven times. Each of the three planned mechanical tests (tensile, bending and impact) then required the manufacturing of 147 specimens. As a result, the total number of 441 STL models of specimens were loaded into Magics 24.1 software (Materialise NV, Leuven, Belgium) with a dedicated nesting module. For successful printing, the nesting algorithm was set to keep a 5 mm distance between each specimen. Moreover, it was defined to keep a similar slice area within each processed layer. The outcome of nesting process is shown in Figure 8. The final height of the print job was 317.4 mm and the packing density (i.e., the ratio of volume occupied by bodies of specimens to the whole available build job volume) was 6.93%.

Figure 9 shows a slice area distribution graph. The majority of the slices have an area in the range of 60–100 cm^2^. Thanks to this, there should be no major changes in heat intensity throughout the layers, whereby it is possible to consider that only the orientation of the specimens influences their mechanical properties. Considerable peaks and valleys in the slice distribution graph usually lead to uneven heat traces and, therefore, the products may be either influenced mechanically or distorted [29].

### 2.4. MJF Process Setup

The MJF printing process was conducted on a HP MJF 5200 machine with an effective building volume of 380 × 284 × 380 mm. The machine is optimised for a layer thickness of 0.08 mm, and may achieve a theoretical building rate of up to 5058 cm^3^/h. The resolution of the printing process on the XY plane is 1200 dpi. In comparison with other additive technologies, MJF excels in many aspects of the process. First of all, no support structures are needed and therefore, it is possible to manufacture very complex shapes. Moreover, a lower used/fresh mixing ratio may be used. For this study, a ratio of 80/20 (80% used and 20% fresh PP powder) was used. In comparison, a typical mixing ratio of used and fresh powder for the SLS technology is 70/30 or 50/50.

### 2.5. Measurement of Mechanical Properties

The Charpy test was performed according to ISO 179-1:2010. The standardised specimens contain a type A notch which in our case was cut after the printing process using a dedicated device. The distance of supports was 62 mm and the width under the notch was 8 mm (Figure 10).

The tensile test was performed according to ASTM D638-14 using a TiraTest universal testing frame. For the further calculation of stress values, each specimen cross-section was measured using Mitutoyo QuantuMike 293-140-30 micrometer (Mitutoyo Corporation, Kawasaki, Kanagawa, Japan). Generally, the measured dimensions were found to be larger than the nominal ones. Highest and lowest deviations were +0.32 mm and −0.07 mm, respectively. Elongation of the specimen was measured using an MFL 800-B extensometer (MF Mess- & Feinwerktechnik GmbH, Velbert, Germany) with an initial gage length of 25 mm. Each specimen was tested under the load ratio of 50 mm/min until failure.

Bending tests were performed according to ISO 178. The length between the supports was 62 mm. The ambient temperature was 23 °C ± 2 °C, and the relative humidity was 50 ± 5%, measured according to ISO 291. The samples were preconditioned for 16 h under these ambient conditions.

Individual data obtained from mechanical tests were first evaluated and displayed using box and whisker plots. For the direct comparison of different specimen configurations, the heatmaps were created. The difference within measures given by different printing orientations was analysed by the Kruskal–Willis test with a significance level of 95%.

## 3. Results

### 3.1. Charpy

The toughness of the tested samples was significantly lower for the specimen orientation of 001, followed by a toughness value for the orientation of 111. The orientation of 110 showed significant differences between its 0°, 45° and 90° configurations, as can be seen in Figure 11. The increased variation in toughness was observed for all the orientation configurations, except in the case of 110 and 111.

### 3.2. Tensile Tests

Young’s modulus $E_{t}$ in the orientation 001–00° significantly differs from most of the other orientation configurations. The significant difference was observed for the pair 011–45° and 011–90°. The variations increased in the case of orientation 111, but also in the other configurations (001, 010, 110), as can be seen in Figure 12. The ultimate strength $R_{m}$ was significantly different for configurations 001–00° and 001–90°. A significant difference between specimens built under 0°, 45° and 90° was found for configurations 010 and 100. The high variations were found for the configuration 001–45°.

The ultimate strain $\varepsilon_{m}$ is shown in Figure 13. In comparison with other orientations, the most different value was found for configuration 010–00°. A high variance was found for configurations 011–90° and 111–90°. The tensile strain $\varepsilon_{b}$ was significantly different for configuration 001. The highest variance was found for 010–45° and 100–90°.

### 3.3. Bending Test

The results of the bending tests are shown in Figure 14. The flexural strength is highest for orientation 111–90°, while it is the lowest for 101–00°. The most significant difference was found for the orientation 101–90°. The highest variance was found for 100–90°. Flexural modulus was the highest for orientation 011–90° and lowest for 101–00°. The highest variance was found for 001–00°. The most significant difference for this quantity was found for orientation 011–90°.

## 4. Discussion

The thermal stability and morphology of the PP powder of different mixing ratios (new, used and mixed) are very similar and hence, the MJF process does not significantly influence the structural and thermal properties of the PP powder. This implies that PP powder may be repeatably used under certain mixing ratios (we used the ratio of 80/20% recommended by a manufacturer).

In view of the different orientations and building directions of the tested specimens, the resultant mechanical properties of the PP specimens were significantly scattered. In the author’s opinion, the broad scattering in the mechanical properties is given mainly by three factors. The first factor represents the influence of the surface area in a particular layer. We hypothesise that printing specimens with a relatively small layer area (this corresponds to orientation 001, where the main sample axis corresponds to the Z axis of the printer) excessively overheats the printing material and hence, the resultant mechanical properties may be degraded to some extent. The second factor may be that the samples were printed with different spatial locations in the building space (for example, close to walls and corners), which may cause an additional fluctuation in the mechanical properties, as the cooling effect is not homogeneous in the building space. The parts that are in the centre of the print space gradually cool linearly, whereas the parts that are in the corners or in the lower or upper part of the printing space will cool faster due to the large heat dissipation through the sidewall or bottom plate. In the upper part, there is a free space where there might be a large temperature drop between the temperature of the parts and the ambient temperature. The described considerations related to the second factor should be verified first before any conclusions can be made.

The third factor may be the anisotropy in the mechanical properties induced by the “layer-by-layer” printing process. It is reasonable to expect that loading the specimens transversely to the orientation of the printing layer may lead to lower mechanical properties, which is a well-known effect of the additive manufacturing process. Our assumptions seem supported by the results, which show that (apart from the ultimate strain), the Young’s modulus, ultimate strength, toughness and tensile strain are systematically low for orientation 001. The high ultimate strain for orientation 001 reflects the fact that the samples are more ductile with a lower load resistance. The exceptionally low tensile strain value of samples printed with orientation 001 may be explained by the fact that passing the ultimate strain, the tensile stress acting on the layers is excessive due to the layer area being the smallest for orientation 001, and hence, the cohesive cracking quickly developed.

Orientation 010 provided the highest area of the printed layer. The elasticity and ultimate strength were significantly improved for orientation 010 compared to orientation 001. The samples printed with orientation 010 were more brittle, but with a high tensile strain indicating that the interlayer resistance to crack propagation was much better (given also by the fact that the thickness of the layer was lower, and hence, more layers were able to better bear the load).

In order to optimally use the MJF printing device, it was necessary to minimise the printing height and maximise the packing density (optimally in the range of 6–11%). Below 6%, the cost-effectiveness of the MJF quickly decreases, as the material waste is enormous.

Compared to the standard injection moulding method and SLS, the samples produced by MJF are generally more rigid. The lowest ultimate tensile strength of 27 MPa was found for orientation 001–90, and in [13], it was only 19.9 MPa. The lowest Young’s modulus was 1200 MPa for orientation 011–90°, which is more than 100% higher than the value of 599 MPa found in [13]. Comparing the strain at breaking $\varepsilon_{b}$, our samples broke at 23% elongation (orientation 110–90°), which is much less than in [13] (122%). The Young’s modulus in bending was slightly smaller (the highest value was 1165 MPa for orientation 011–90°) than in tension, but still much higher regarding the SLS. The bending ultimate strength was slightly higher than the tensile (the maximum was 35.3 MPa for orientation 111–90°), but again, compared to SLS, the values are still much higher. Compared to the injection moulding, the mechanical properties are even more different—for details, see [13].

This study did not take into account the different mixing ratios and their influence on the mechanical properties of PP samples, but these issues are a priority list and will be part of a future study.

## 5. Conclusions

The ability of MJF printing technology to accept PP powder opens up new possibilities in the decision-making process in material/structural engineering. The efficiency of MJF and the unique mechanical and chemical resistance of PP allows many products to be rapidly produced at a larger scale. This study demonstrated the following:The MJF process has no significant impact on the structural and thermal properties of PP powder, and hence, can be reused,The sample orientation has a significant influence on the impact mechanical properties of PP powder,The sample orientation has a significant impact on the tensile mechanical properties of PP powder.

## Figures and Tables

**Figure 1 materials-14-02165-f001:**
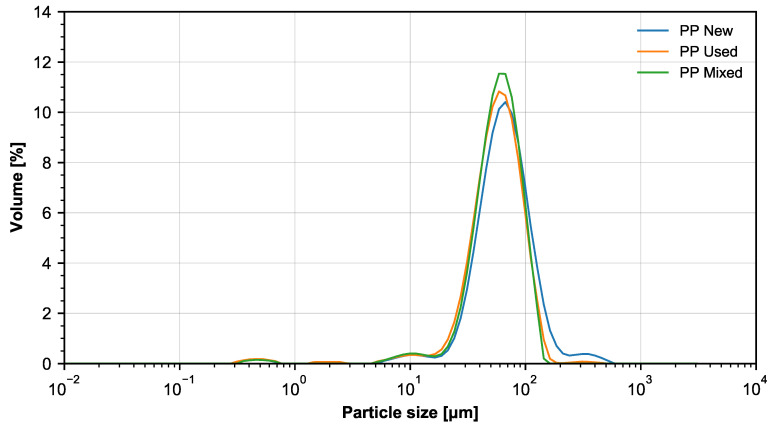
The histograms of the size distribution of new, used and mixed PP powder.

**Figure 2 materials-14-02165-f002:**
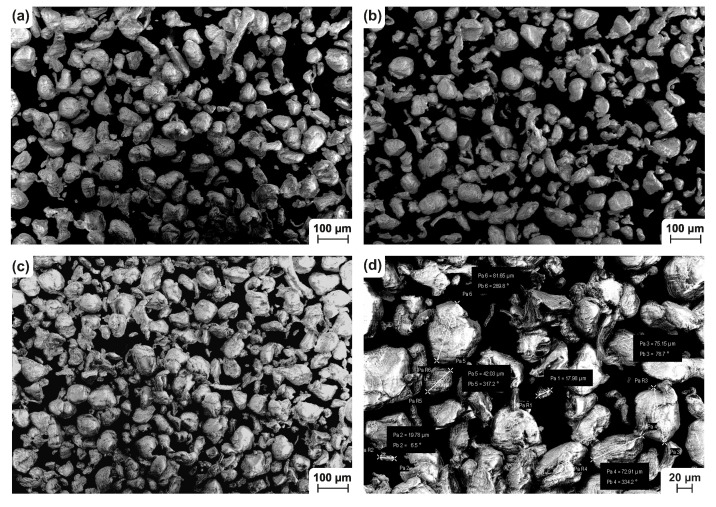
SEM images of PP powder: (**a**) new; (**b**) used; (**c**) and (**d**) mixed.

**Figure 3 materials-14-02165-f003:**
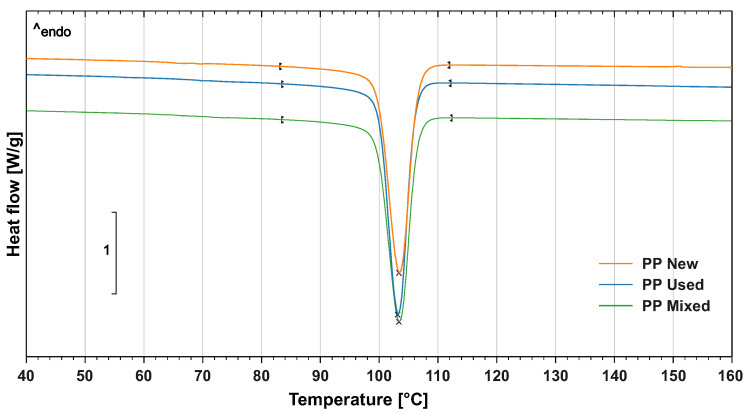
Crystallinity point of (new, used, mixed) PP powders.

**Figure 4 materials-14-02165-f004:**
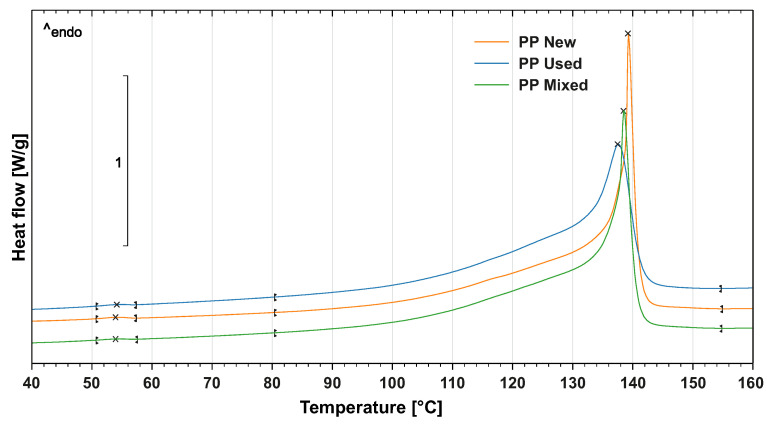
Melting point of (new, used, mixed) PP powders.

**Figure 5 materials-14-02165-f005:**
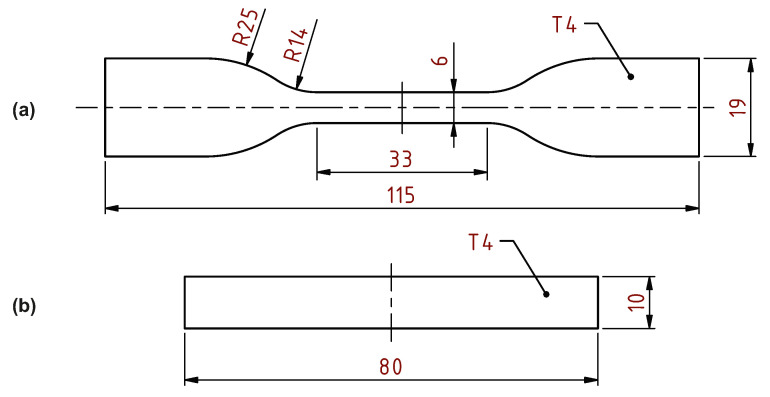
Specimens for (**a**) tensile; and (**b**) bending and impact tests.

**Figure 6 materials-14-02165-f006:**
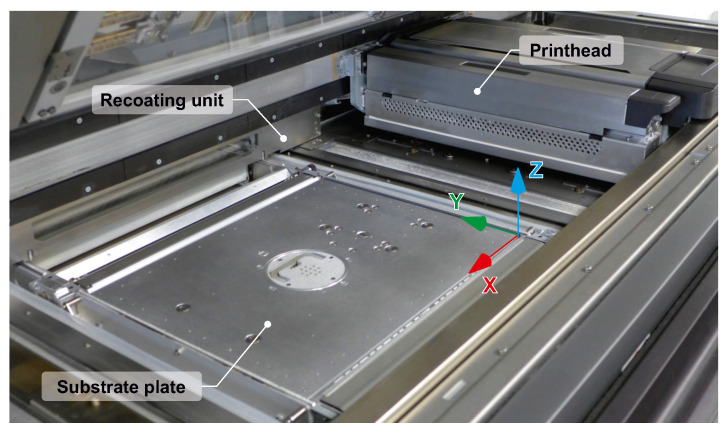
Coordinate system displayed in the build space of the HP machine.

**Figure 7 materials-14-02165-f007:**
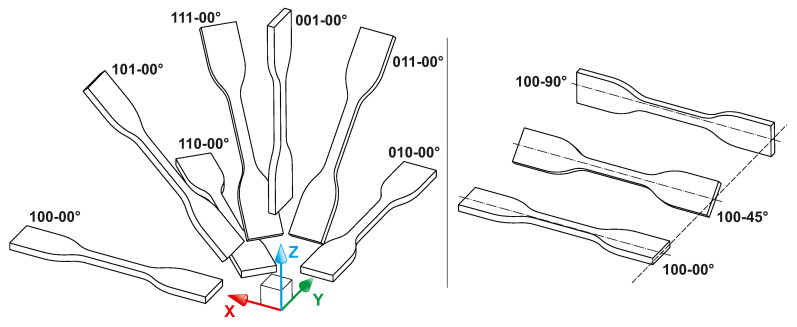
Position of individual specimens.

**Figure 8 materials-14-02165-f008:**
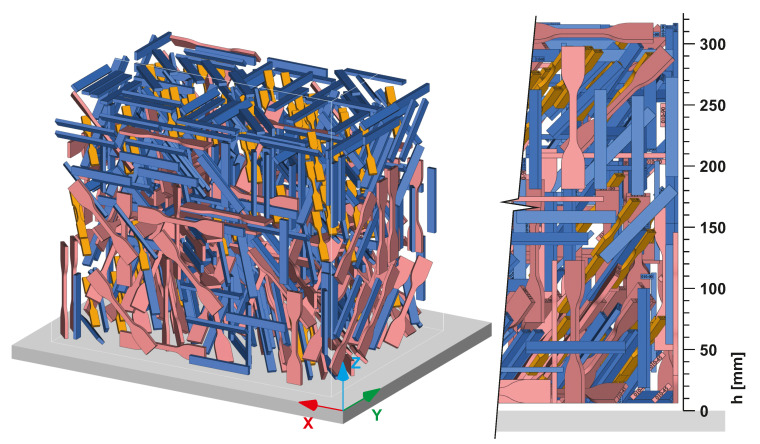
Overview of print batch with all the specified polypropylene specimens.

**Figure 9 materials-14-02165-f009:**
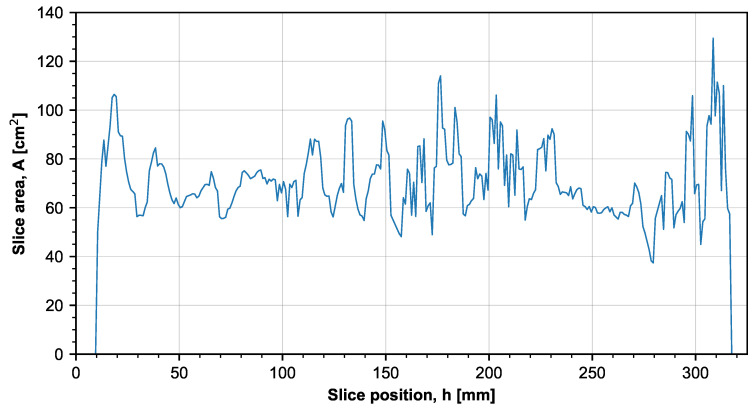
Slice area on individual layers of the build job.

**Figure 10 materials-14-02165-f010:**
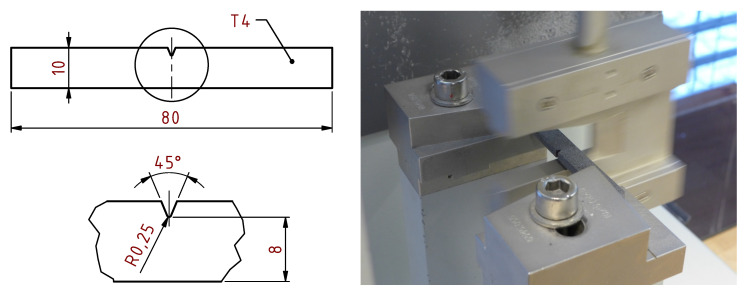
Notched specimen for impact testing.

**Figure 11 materials-14-02165-f011:**
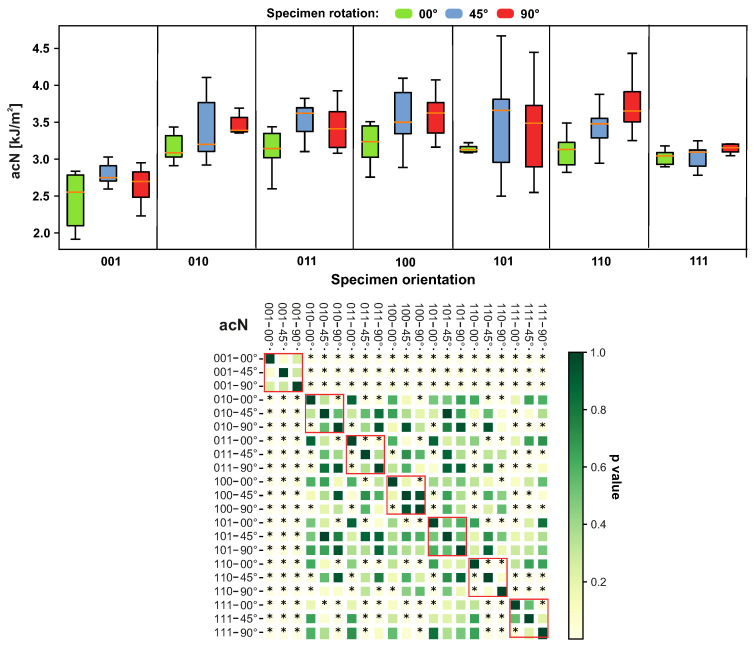
Impact of different sample orientations on the impact strength acN. The red border in heatmap highlights specimens with equal orientation. The members marked as * express a statistically significant difference.

**Figure 12 materials-14-02165-f012:**
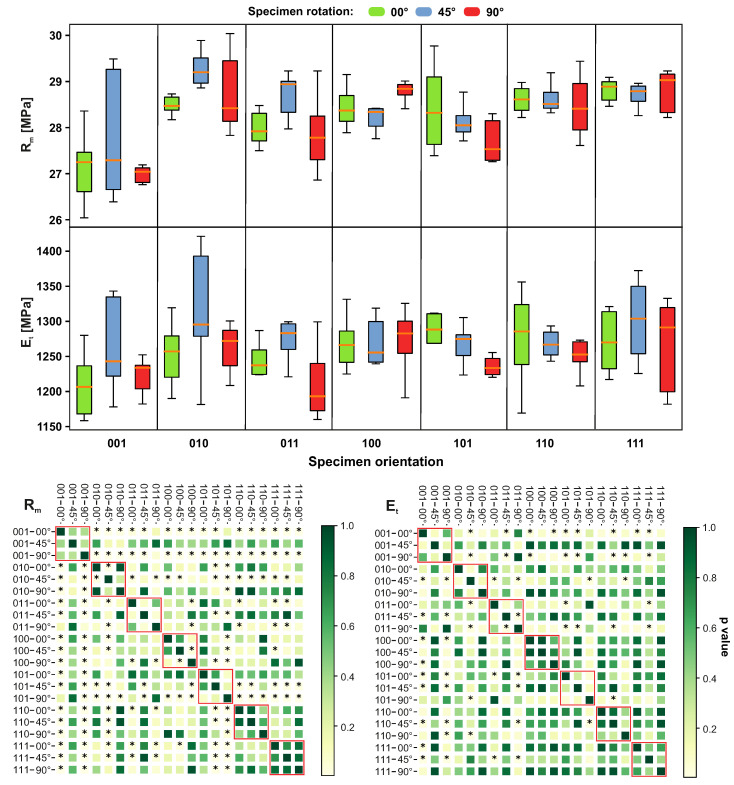
Impact of different sample orientations on Young’s modulus ($E_{t}$) and ultimate tensile strength ($R_{m}$) values. The red border in the heatmap highlights specimens with equal orientation. The members marked as * express statistically significant differences.

**Figure 13 materials-14-02165-f013:**
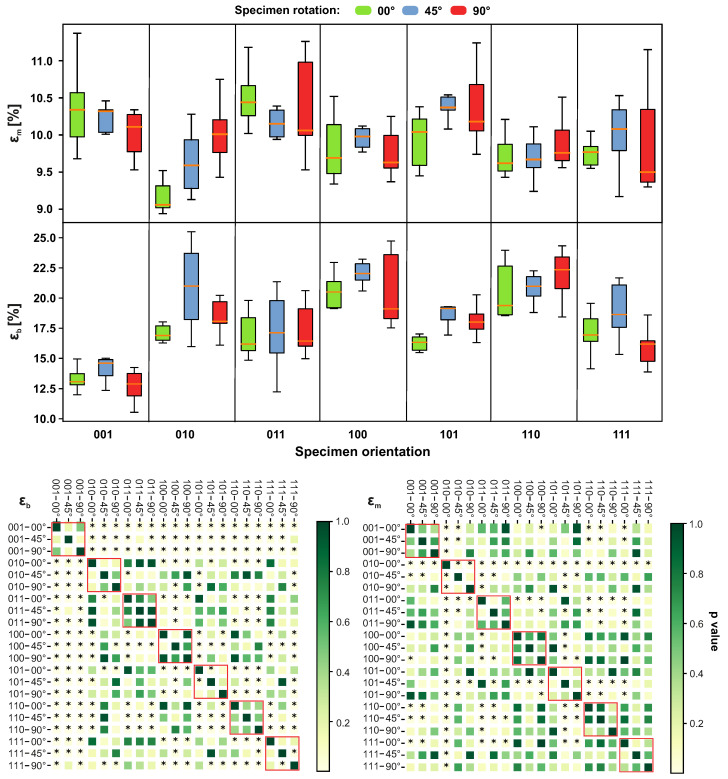
Impact of different sample orientations on the ultimate strain ($\varepsilon_{m}$) and strain at break ($\varepsilon_{b}$) values. The red border in the heatmap highlights specimens with equal orientation. The members marked as * express a statistically significant difference.

**Figure 14 materials-14-02165-f014:**
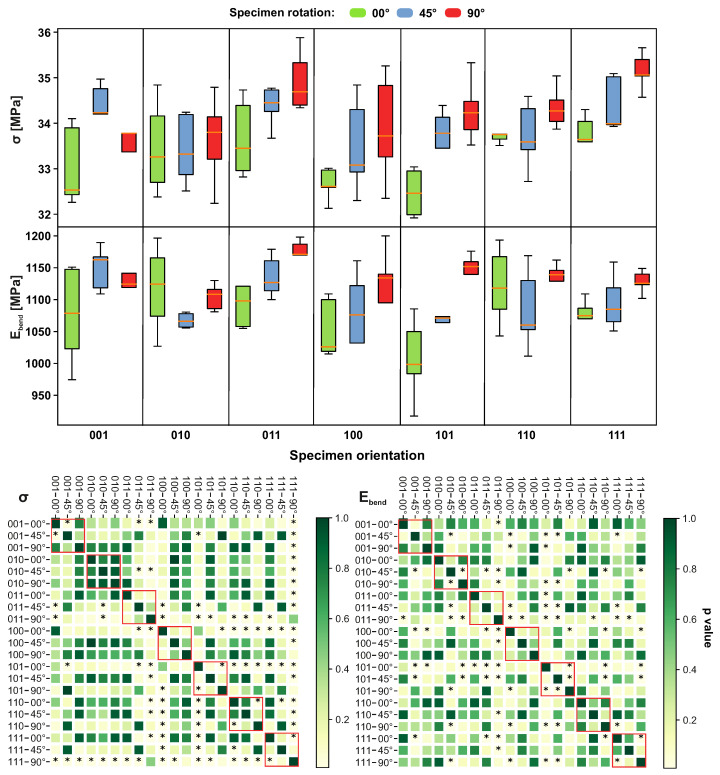
The impact of different sample orientations on flexural strength (σ) and flexural modulus ($E_{bend}$) values. The red border in the heatmap highlights specimens with equal orientation. The members marked as * express a statistically significant difference.

**Table 1 materials-14-02165-t001:** Percentile diameter distribution of powder size for new, used and mixed PP powder.

	D10 (m)	D50 (m)	D90 (m)
PP new	35.5	69.0	130.0
PP used	31.2	61.6	108.0
PP mixed	33.3	62.8	105.0

**Table 2 materials-14-02165-t002:** Parameters of SEM analysis for PP powder morphology analysis.

Parameter	Value
Accelerating voltage	15 kV
Aperture	12 mm
Distance of specimen	8.2–8.5 mm
Scan speed	5
Detector	SE2
Magnification	100×, 250×

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
