# Peer review of "Mechanical Properties of Polypropylene: Additive Manufacturing by Multi Jet Fusion Technology"

_materials, 2021, doi:10.3390/ma14092165_

Round 1

Reviewer 1 Report

Few Comments and Suggestions:

  • page 5 about the introduction of "packing density" It would help to define it;
  • page 5: "Considerable peaks and valleys ..... " this assumption sounds general and without reference;
  • page 6 - row 112: 

    At this point of the activity description I suggest to include some details on the geometric characteristics of the obtained specimens in terms of real obtained dimensions;

  • page 9 row 184 the reported  consideration should be integrated with reference/measured temperature values or temperature maps of the printing chamber. About this point it would be also interesting to have a picture of the printed work with some considerations

Reviewer 2 Report

The article about mechanical properties of polypropylene manufactured by multi Jet fusion technology is well written and after addressing a few details it is suitable for being published in materials. More in details

I would omit table 2 since the details for the image acquisition are already provided in the text,

Line 92 and 93 . The authors wrote we obtained 21 unique specific specimens and each combination was printed seven times.  So there should be 7x21 = 147  samples. They also wrote 441 specimen were prepared. Can you please elaborate further?

Line 127 weight of the samples 6-9 g can you please provide the model of the TGA device and DSC device ? generally the size of the sample is quite smaller in comparison with the sample used by the authors (mg and not g)  Also can you please specify the number of replicates performed for the DSC analysis (quite sensitive) and TG ?

Can the authors please add a few more details about the parameters used for the  bootstrapping process?

The figures are very useful in explaining the results. The heatmap are a very good choice to show lots of information in the same plot. Eventually the authors are suggested to explore (for further work or just for curiosity) the suitability of the use of jiter plot (strip plot) instead of box and whisker plot. While dealing with samples in the hundreds they have the advantage to visually show the distribution for all the sample without relying on summarizing them

Reviewer 3 Report

The study describes the research of, inter alia, mechanical properties of prints made in the MJF technique. The presented scope of research and analyzes is very extensive.
The reviewer's doubts are raised by the fact that all samples are printed in one process. This resulted in a random arrangement of samples in a given orientation in the working space of the machine. Such random arrangement introduces additional problems in the analysis of the influence of the orientation of the sample geometry in relation to the printing direction.
The authors wanted to demonstrate in one experiment the aspects related to the influence of the sample arrangement on the mechanical properties and technological aspects (e.g. process efficiency). Such a connection makes it difficult to unequivocally assess the results of the research, which the authors themselves note.
However, the obtained results can be used by other researchers, as well as by people who design processes for manufacturing plastic components.
Suggestion:
Change the statement "similar heat intensity" (line 101) because the differences in the filling of a given layer reach an average value of about 40%. Therefore, a similar term is not appropriate. The authors themselves, when discussing the results, pay attention to this fact (line 175).

Reviewer 4 Report

This is an extremely interesting work on the mechanical characterization of materials used for the MJF 3d printing technique.

The topic is very innovative and the study is well followed and designed. Some criticisms are however present:

An opening sentence in the abstract section should be added on the possible fields of application of CAD / CAM technology in the medical field

-The abstract section must be completely rewritten because it is unclear. In particular, it should be organized, after the general part, in objectives of this study, methods used, results without numerical values ​​and clinical significance which is completely lacking. The individual sections do not have to be indicated in the paragraphs

-The general part of the introduction, before talking specifically about the MJF, should be reserved for an examination of the main methods of 3d and 4d printing, differentiating additive techniques from the classic subtractive ones

-In the final part of the introduction, the null hypotheses of the study must be inserted, which must be refuted at the end in the light of the results obtained

- the materials and methods section is well structured and complete but must be arranged for the test performed and not for theoretical discourse with specific subparagraphs. No results should be reported in this section

-The Conclusion section should also be changed. In the initial part, some general considerations must be added on the possible fields of application of CAD / CAM technology in the medical field. In this regard, I recommend inserting the following scientific works in the reference section which could be of help to the reader in understanding these areas:

-Lancellotta V, Pagano S, Tagliaferri L, Piergentini M, Ricci A, Montecchiani S, Saldi S, Chierchini S, Cianetti S, Valentini V, Kovács G, Aristei C. Individual 3-dimensional printed mold for treating hard palate carcinoma with brachytherapy: A clinical report. J Prosthet Dent. 2019 Apr; 121 (4): 690-693. doi: 10.1016 / j.prosdent.2018.06.016. Epub 2018 Nov 30. PMID: 30503148.

-Di Stadio A, Gambacorta V, Cristi MC, Ralli M, Pindozzi S, Tassi L, Greco A, Lomurno G, Giampietro R. The use of povidone-iodine and sugar solution in surgical wound dehiscence in the head and neck following radio-chemotherapy. Int Wound J. 2019 Aug;16(4):909-915. doi: 10.1111/iwj.13118. Epub 2019 Apr 10. PMID: 30972904.
